# A Strategy for the Recovery of Raw Ewe’s Milk Microbiodiversity to Develop Natural Starter Cultures for Traditional Foods

**DOI:** 10.3390/microorganisms11040823

**Published:** 2023-03-23

**Authors:** Luigi Chessa, Antonio Paba, Ilaria Dupré, Elisabetta Daga, Maria Carmen Fozzi, Roberta Comunian

**Affiliations:** Agris Sardegna, Servizio per la Ricerca nelle Produzioni Animali, Associated Member of the JRU MIRRI-IT, Loc. Bonassai SS 291 km 18.600, 07100 Sassari, Italy; lchessa@agrisricerca.it (L.C.); apaba@agrisricerca.it (A.P.); idupre@agrisricerca.it (I.D.); edaga@agrisricerca.it (E.D.); mfozzi@agrisricerca.it (M.C.F.)

**Keywords:** natural starter cultures, ewe’s milk, biodiversity recovery, lactic acid bacteria

## Abstract

Commercial starter cultures, composed of high concentrations of a few species/strains of lactic acid bacteria (LAB), selected based on their strong technological aptitudes, have been developed to easily and safely carry out food fermentations. Frequently applied to industrial productions, selected starter LAB easily become the dominant microbiota of products, causing a dramatic decrease in biodiversity. On the contrary, natural starter cultures, which usually characterize the most typical and Protected Designation of Origin (PDO) food products, are constituted by a multitude and an indefinite number of LAB species and strains, both starter and nonstarter, thus contributing to preserving microbial biodiversity. However, their use is not risk-free since, if obtained without heat treatment application, natural cultures can contain, together with useful, also spoilage microorganisms or pathogens that could be allowed to multiply during fermentation. In the present study, an innovative method for the production of a natural starter culture directly from raw ewe’s milk, inhibiting the growth of spoilage and potentially pathogenic bacteria without applying any heat treatment, was described. The culture developed show a good degree of microbial biodiversity and could be applied to both artisanal and industrial scales, guaranteeing safety, quality constancy, technological performance reproducibility, preserving biodiversity and peculiar sensory characteristics, usually linked to traditional products, while overcoming the problems associated with the daily propagation of natural cultures.

## 1. Introduction

The ancient cheesemaking technologies involved the use, as starter, of previously heat-treated milk incubated to favour the development of useful microbiota (*latto-innesto*), whey or *scotta* (the residual whey of ricotta cheese manufacturing), coming from a successful production, after overnight fermentation (*siero*/*scotta-innesto*) [1,2]. More recently, there has been a need to extend fermentation at an industrial scale to standardise the process, which requires the use of starter cultures with adequate technological performances. Proper development of selected starters represents a convenient solution to easily and safely carry out the fermentation when the concentration of the microbiota colonizing the production environment and the raw material itself turns out to be inadequate or natural starter cultures are difficult to obtain and manage [1,3,4].

In the development of a starter culture for cheesemaking, two ways can be mainly pursued. The first is the selection of a low and defined number of species/strains, mostly lactic acid bacteria (LAB), based on their strong aptitude to fulfil the biochemical processes required by each production technology and their suitability to be grown in the laboratory [5]. Although the selection of specific microorganisms allows the standardisation of the fermentation, it cannot fully reflect the diversity of fermented food varieties [6], especially the typical and Protected Designation of Origin (PDO) cheeses. The second method concerns the reproduction of autochthonous biodiverse natural cultures where an undefined number of species and strains, starters and nonstarters (but crucial during the whole fermentation and ripening of food) coexisting in equilibrium, derived from an old backslopping practice and selective heat pressure [5,7]. Both choices have advantages and disadvantages. On one hand, selected cultures, because of and despite their high technological efficiency, easily become the dominant microbiota of the product, causing a dramatic decrease in microbial biodiversity and loss of peculiar sensory characteristics of fermented food. In fact, this kind of culture is widely applied, at high concentration, to industrial-level productions that do not possess geographic niches and typicity. On the contrary, natural microbial communities have a strain composition that is not reproducible in any place other than that of their origin, contributing to preserving microbial biodiversity and enriching products with peculiar sensory features that bind them to the territory. Indeed, autochthonous natural starter cultures, better adapted to the raw material to be processed [8,9], usually characterize the most typical and high-quality agrifood products and are more phage-resistant than the selected ones [10]. However, their technological performance cannot be standardized, and their use cannot be risk-free since, together with useful autochthonous microorganisms, even pathogens or spoilage could be potentially inoculated and allowed to contaminate the product. To overcome this problem, raw milk is generally subjected to heat treatment to possibly eliminate spoilage and pathogenic bacteria [11]. However, heat treatments cause a loss of the original microbial biodiversity in milk that cannot be restored by the addition of selected starters. Conversely, the use of autochthonous, natural, biodiverse starters can help maintain microbial milk biodiversity and preserve the typicity of dairy products. Besides, natural cultures contain, over starter LAB (SLAB), even non-starter LAB (NSLAB) that could positively influence human gut composition and have a probiotic potential, improving consumers’ health [12,13]. Anyway, since these kinds of starter cultures are intended for addition to food, they can contain strains not included in the list of QPS (qualified presumption of safety) or GRAS (generally recognized as safe) species, and a safety evaluation should be performed according to the EFSA [14] or FDA [15] suggestions.

The aim of the present study was to set up a novel method to obtain a natural starter culture for cheesemaking directly from raw milk, avoiding the usual practices involving heat milk treatments and/or selection by production technology. Furthermore, this method of bypassing milk thermisation and avoiding the integration with commercial starters is also aimed at recovering most of the natural microbial biodiversity of milk.

## 2. Materials and Methods

### 2.1. Experimental Design

The raw milk was collected from a sheep farm located in the centre of Sardinia (Italy), investigated for its microbial composition, and an aliquot was used for the preparation of the natural starter culture following the method described below. Then, the starter culture was tested in different media suitable for its propagation and in milk to assess its technological properties. Microbial counts and molecular fingerprints, by rep-(GTG)_5_ PCR, were performed to monitor the possible shifts of the whole microbial community during propagation.

A graphical description of the experimental design is shown in Figure 1.

### 2.2. Milk Sampling and Starter Culture Preparation

The raw ewe milk (RM) from two subsequent milkings, from the morning and the prior evening, were kept refrigerated in the same tank, aseptically sampled at the ewe farm, immediately transferred, under refrigeration, to the laboratory, supplemented with 15% sterile glycerol, divided into subsamples, and stored at −80 °C.

Before the preparation of the starter culture, an aliquot of RM was thawed at 37 °C, and its composition in dairy, anti-dairy, and hygiene-related microbial groups was investigated by microbial counts on different culture media (Section 2.4 Microbial Counts). An aliquot of thawed milk was also plated onto acidified agarose ewe whey dishes. Operatively, powder ewe whey (W) (Demi90, Alimenta S.r.l., Cagliari, Italy) was dissolved in deionised water (73.7 g/L) supplemented with microbiological agar (Microbiol, Cagliari, Italy) (15 g/L), stirred for 20 min until dissolved, adjusted to pH 5.5, 6.0, and 6.5, autoclaved at 105 °C for 20 min (pH decreased to 5.3, 5.8, and 6.2, respectively) and plated onto petri dishes. Next, 1 mL RM aliquots were spread on the agar whey 120 mm diameter Petri dishes (at pH 5.3, 5.8, and 6.2) and incubated at 37 °C for 24 h anaerobically. After incubation, the entire agar content of each plate was aseptically transferred into a glass bottle containing 200 mL of sterile whey broth at pH 6.7 and incubated at 37 °C for 24 h. After incubation, the bottle’s contents were carefully mixed by hand, and 20 mL of each were used for the inoculation of three distinct bottles containing 180 mL of sterile (105 °C for 20 min) ewe’s milk. The inoculated milk was incubated at 37 °C and the acidification was continuously monitored and recorded by an eight-channel Liquiline CM448 transmitter (Endress+Hauser, Gerlingen, Germany) coupled with Field Data Manager Software (v. 1.03, Endress+Hauser). After coagulation, aliquots of fermented milk (FM) from each of the three theses (whey agar dishes at pH 5.3, 5.8, and 6.2) were tested for the presence of coliforms (see Section 2.4 Microbial Counts), and the remaining FM was supplemented with 15% sterile glycerol and stored at −80 °C.

The natural starter culture (NSC) achieved in this study was obtained by the inoculation of FM (0.5% *v*/*v*) in sterile ewe milk and incubating at 37 °C until milk coagulation. The acidification curve was monitored continuously, and the development of the microbial consortia at the end of acidification was investigated. NSC was then supplemented with 15% sterile glycerol, divided into aliquots, and stored at −80 °C.

The protocol described to obtain the natural starter culture (NSC) was carried out with three distinct replicates.

### 2.3. Starter Culture Characterisation

#### 2.3.1. Acidification Ability

During ewe’s milk incubation at 37 °C to obtain FM and NSC, acidification was monitored and recorded continuously by an eight-channel Liquiline CM448 transmitter (Endress+Hauser, Gerlingen, Germany) coupled with Field Data Manager Software (v. 1.03, Endress+Hauser). Moreover, the acidification ability of NSC was also tested in milk (M), in whey (W), and in W ultrafiltered (WU) with the Hydrosart Module for Sartocon Slice (Sartorius Stedim Biotech, Gottinga, Germany) at 30 kD. The acidification was also tested in W ultrafiltered and supplemented with 8.2 g/L of bacteriological peptone (Microbiol) (WU + P) to restore the original nitrogen concentration in W.

To evaluate the technological performances suitable for cheesemaking, milk coagulation ability and kinetic curves of acidification were also tested in triplicate and compared (Section 2.5 Statistical Analysis).

#### 2.3.2. Microbial Composition

The natural starter culture (NSC) was characterised for its microbial composition in alive cells belonging to thermophilic cocci and lactobacilli, mesophilic cocci and lactobacilli, enterococci, citrate-fermenting bacteria, staphylococci, and coliforms (Section 2.4 Microbial Counts).

From each set of agar plates used for the enumeration of bacteria in NSC, 10 colonies were picked up from the lowest countable dilution and observed using an AxioPhot optic microscope (Carl Zeiss, Oberkochen, Germany) equipped with Objective N-Achroplan 100×/1.25 Oil Ph3 M27. The different shapes and arrangements of bacterial cells were determined.

Total-DNA extraction from NSC was performed following the protocol described by Paba et al. [16], and genus/species-specific PCR analysis was carried out for the detection of the microbial genus/species listed in Table 1.

#### 2.3.3. Molecular Fingerprint 

Microbial consortia fingerprints of NSC and of NSC grown in M, W, WU, and WU + P were obtained from total DNA by rep-(GTG)_5_ PCR analysis using FTA^®^ Disc for DNA analysis (GE Healthcare, Chicago, IL, USA) directly as a template, as described by Chessa et al. [25]. PCR products were separated on an agarose gel (1.8% *w*/*v*) with SYBR SAFE 1× (Invitrogen-Fisher Scientific, Rodano, Italy), at 100 V (222 V/h) in Tris-acetate buffer. Gel images were acquired with the UV transilluminator FireReader V4 (UVITec, Warwickshire, UK) in *tiff* format, then elaborated as described in Section 2.5
*Statistical Analysis*.

### 2.4. Microbial Counts

The concentration of dairy, anti-dairy, and hygienic-related microbial groups in RM, FM, NSC, and of NSC grown in M, W, WU, and WU + P was evaluated. 

Aliquots of 1 mL of each serial ten-fold dilution in sterile saline solution (0.89% *w*/*v* NaCl) were seeded on a duplicate set of agar plates containing specific elective media for each microbial group. Thermophilic cocci were enumerated on M17 agar (Microbiol) at 37 °C for 48 h anaerobically using Oxoid™ AnaeroGen™ (Thermo Fisher Scientific, Waltham, MA, USA); thermophilic lactobacilli on MRS agar pH 5.4 (Microbiol) at 45 °C for 48 h anaerobically; mesophilic cocci on M17 agar (Microbiol) at 30 °C for 48 h anaerobically; mesophilic lactobacilli on FH agar [26] at 37 °C for 72 h anaerobically; enterococci on KAA (Microbiol) at 42 °C for 48 h aerobically; moulds and yeasts on MEA (Biokar Diagnostic, Allonne, France), supplemented with Chloramphenicol 100 mg/L at 25 °C for 3–5 days aerobically; citrate-fermenting bacteria on modified MRS agar [27] at 37 °C for 72 h anaerobically; staphylococci on MSA (Biokar Diagnostic) at 37 °C for 24 h aerobically; and coliforms on VRBA mug (Biolife Italiana, Monza, Italy) at 37 °C for 18 h aerobically.

The presence of coliform colonies of faecal origin in the VRBA mug was checked by a Wood UV lamp. A catalase test for staphylococcal colonies, grown in MSA dishes, was performed.

Microbial counts were expressed as average values ± standard deviation (SD) in Log CFU/mL or Log CFU/g.

### 2.5. Statistical Analysis

The microbial growth of NSC inoculated in M, W, WU, and WU + P was statistically compared by one-way analysis of variance (ANOVA). Differences among the individual means of each microbial group in the four substrates were compared by the Tukey–Kramer post hoc test (*P* < 0.05) using SPSS Statistics (v. 21.0; IBM Corp., Armonk, NY, USA).

In parallel, three replicates of kinetic curves of acidification of NSC grown in M, W, WU, and WU + P were processed by Prism (v. 7; GraphPad Software, La Jolla, CA, USA), and linear regression was used for the curves’ slopes calculation and comparison by the Student *t*-test (*P* < 0.05).

Molecular (GTG)_5_ rep-PCR fingerprint profiles were elaborated by BioNumerics (v. 5.0; Applied Maths, Sint-Martens-Latem, Belgium). Cluster analysis was performed by Pearson’s correlation index through the unweighted pair group method using arithmetic averages (UPGMA).

## 3. Results

### 3.1. Starter Culture Preparation

The microbiota of raw milk (RM), observed by a plate count assay, consisted mainly of cocci-shaped bacteria. In particular, thermophilic cocci were 5.04 Log CFU/mL and mesophilic cocci were 5.97 Log CFU/mL, whereas lactobacilli, both thermophilic and mesophilic, were 1–2 Log lower than cocci: 3.97 and 3.65 Log CFU/mL, respectively (Figure 2). Enterococci were 4.63 Log CFU/mL, yeasts were 3.86, citrate-fermenting bacteria were 0.49, and staphylococci (not further investigated) were 5.06. Coliform bacteria were 4.76 Log CFU/mL, highlighting the poor hygienic quality of the raw milk. Indeed, half of the coliforms turned out to be of faecal origin.

To inhibit the growth of spoilage bacteria, RM whey agar dishes (pH adjusted at 5.3, 5.8, or 6.2), after incubation, were transferred into a bottle of sterile whey broth (pH 6.7) and incubated again for 24 h (Figure 1). The microbial consortia developed evidenced the absence of coliforms in all three methods investigated (i.e., dishes at pH 5.3, 5.8, and 6.2). The thesis at pH 6.2 allowed the inhibition of coliforms while likely limiting the changing of the microbial consortium of raw milk to a minimum, hence it was considered the most conservative and then brought forward. An aliquot of the incubated whey was used to inoculate sterile milk at 10%, reaching a concentration of 8.9 Log CFU/mL. After incubation, the microbial consortium in fermented milk (FM) was well developed, having a cocci-shaped (both thermophilic and mesophilic) concentration of 8.61 and 8.37 Log CFU/mL, whereas lactobacilli were about 1 Log lower, and enterococci reached 6.34 Log CFU/mL (Figure 2). The important goal achieved by this step was the absence of coliforms and staphylococci, which were present in RM but not detected in FM. Moreover, the microbial consortium revealed good acidification ability, reaching pH 5.0 after 6 h of incubation, and it was then used for the inoculation (6.9 CFU/mL) of sterile ewe milk to obtain the starter culture (NSC), through incubation until coagulation (pH 5.0, 440 min). Overall, the NSC viable counts were slightly higher (about 0.7 Log CFU/mL) than FM counts, while enterococci increased 1.2 Log CFU/mL (Figure 2). As expected, coliforms and staphylococci were not detected. This microbial consortium, developed in milk, was considered the natural starter culture (NSC) obtained in this study.

### 3.2. Starter Culture Microbial Characterisation

The natural starter culture (NSC) was characterised for its microbial composition. The concentrations of both thermophilic and mesophilic cocci matched (9.08 Log CFU/mL), whereas presumptive lactobacilli were lower: 7.96 Log CFU/mL for thermophilic and 7.36 Log CFU/mL for mesophilic (Figure 2). Enterococci were found at 7.50 Log CFU/mL, and no staphylococci nor coliforms were detected. Microscopic observation of the microbial colonies grown in M17 and MRS agar media, used for the enumeration of thermophilic and mesophilic cocci and thermophilic lactobacilli in the NSC, revealed the presence of only olive-shaped or round cocci arranged in diplococci in short and long chains. Rod-shaped bacteria, arranged in pairs or short chains, in FH medium were observed.

Molecular analysis by genus/species-specific PCR performed on the total DNA extracted directly from the NSC showed the presence of *Lactococcus lactis*, *Streptococcus gallolyticus* subsp. *macedonicus*, *Lacticaseibacillus paracasei*, *Enterococcus faecium*, and *Enterococcus faecalis*.

### 3.3. Starter Culture Propagation and Technological Performances

The propagation and the technological performances of the NSC in ewe’s milk (M) and in ewe’s whey (with small modifications), another natural medium appropriate for the cultivation of this dairy starter culture, were investigated. The purpose was to find a standard medium suitable to reproduce the natural culture minimising changes in its microbial composition. The NSC was inoculated at 5.9 Log CFU/mL in milk (M), whey (W), W ultrafiltered (WU), and WU supplemented with 8.2 g/L of bacteriological peptone (WU + P) to reconstitute the original protein content present in W. Indeed, ultrafiltration reduced protein concentration from 1% to 0.18% but not lactose.

The concentration of thermophilic cocci and lactobacilli and mesophilic cocci was not statistically different (*P* < 0.05) among NSC and NSC grown in M or in the whey media, except in WU (Figure 2). Mesophilic lactobacilli in M were significantly (*P* < 0.05) lower than in NSC and in the whey media. Enterococci were the only microbial group whose growth was not affected by any of the different media used. Moulds, yeasts, citrate-fermenting bacteria, staphylococci, and coliforms were not detected.

To assess differences in the microbial composition, the molecular fingerprints of the consortia of NSC and NSC grown in M and in the whey media were compared (Figure 3) using a 97.4% similarity cut-off, calculated based on the comparison of three independent (GTG)_5_ rep-fingerprints of each of the consortia investigated. NSC shared a 75.2% similarity with the microbial consortia grown in M and in the whey media. Moreover, the whey media clustered together (97.6% similarity, without substrate-dependent clustering), sharing 95.4% similarity with M.

NSC revealed good technological performances in M at 37 °C, reaching pH 5.2, with coagulation, in 300 min (Figure 4), starting from a microbial concentration of 5.9 Log CFU/mL, the same used for the propagation trials. Distinct acidification performances, depending on the whey media used, were observed (Figure 4). In W and WU, the acidification curves were comparable, with similar *(P* < 0.05) curves’ slopes and final pH after 420 min of incubation (4.93 and 5.06 for W and WU, respectively). In WU + P, NSC reached pH 4.8 rapidly, in 245 min, with a slope significantly (*P* < 0.05) different from W and WU. 

## 4. Discussion

The purpose of this study was the production of a biodiverse natural starter culture directly from raw milk, bypassing any heat treatment, and suitable for cheesemaking applications. This study is a continuum of previous research conducted on natural starter cultures about their propagation, biodiversity, technological performances, and exploitation [2,25,28]. Indeed, half-century preserved natural starter cultures for Pecorino Romano PDO cheese, obtained in the late 1960s from *scotta* (the residual whey of ricotta cheese manufacturing), were reproduced and propagated in the natural medium from which the cultures were obtained (i.e., *scotta*), characterised for their biodiversity, and successfully used for cheese manufacturing [29]. In this study, a natural starter culture was obtained directly from raw ewe’s milk, trying to recover as much as possible of the microbial milk biodiversity, following a different way to the processes carried out at industrial or artisanal scale. Commercial starters consist of single or a few species/strains isolated from milk or food products in the production environment, microbiologically characterised and selected for their technological performances, then grown separately and blended into a final starter ready-to-use with a defined microbial composition. At the artisanal scale, *latto-innesto*, a starter culture obtained by natural fermentation of thermised milk, or *siero*/*scotta-innesto*, are generally used as a microbial inoculum for food fermentation. For the production of *latto-innesto*, raw milk thermisation (55–68 °C) to reduce the number of spoilage bacteria [11], even potential pathogens, is generally applied. However, this practice also has some drawbacks, such as protein denaturation, which can affect cheese sensory quality [6,30,31], and particularly the loss of microbial biodiversity, both on SLAB and NSLAB, which are important for the development of peculiar organoleptic characteristics during ripening [9,12]. In the method described in this study, to preserve and recover as much microbial diversity as possible, no heat treatment to milk was applied to obtain a natural starter culture by removing potentially food-borne pathogens or low-hygienic-related bacteria. Indeed, the raw ewe’s milk used comprised staphylococci (5.06 Log CFU/mL) and coliforms (4.76 Log CFU/mL), half of which were of faecal origin. Despite the low hygiene quality of the raw milk, its cultivation on agarose-whey petri dishes at pH 6.2 was as effective as, but likely less impactful, the lower pHs tested (5.3 and 5.8) in inhibiting the growth of spoilage bacteria, since in the successive steps they were no longer detected. The microbiological quality of RM can affect obtainment of a good starter culture. Though it is not difficult to find a dairy farm able to set up normal good management practices for milk production (starting from animal feeding, through milking, up to milk storage conditions before use), finding raw milk microbiologically too contaminated or too clean can happen. In this case, analysing some RM samples and running the procedure several times could be necessary before finding the most suitable conditions to obtain NSC.

The recovery of microbial biodiversity is a challenge for better sustainability in the microbiological processes of agro-ecosystem products, such as dairy fermentation. Indeed, the use of biodiverse microbial communities in food fermentation can have several advantages from the point of view of better adaptability to the raw material to be processed, flavour production, and high resistance to bacteriophage attacks (in the case of SLAB) [32,33]. A recent study on table olive fermentation demonstrated the efficiency of a natural starter culture with complex and undefined microbial composition (obtained from a previous natural fermentation of the same olive cultivar), compared to a starter composed of two of the best-performing strains isolated from the natural starter and a commercial allochthonous starter [34,35]. The natural starter was more efficacious in olive fermentation, leading also to a safer product, and the study highlighted that autochthonous complex microbial communities coming from the same environment as the raw material to be processed have more adaptability to harsh fermentation conditions, preserving the safety and quality characteristics of naturally fermented olives faster and reducing production costs.

The natural starter culture (NSC) obtained in this study showed good acidification performances coagulating ewe’s milk in 300 min, a time suitable for several cheesemaking productions. A method for the reproduction, at laboratory scale, of NSC using a natural medium (i.e., whey), in a different form (agar instead of liquid), was also investigated. Whey can be considered a suitable substrate for this kind of natural culture growth since it reduces the risk of changes in the microbial composition [25] and can be used to maximize the microbial cell yield, according to Chessa et al. [2]. For the propagation trials of NSC in this study, powder whey was standardly reconstituted and used both unmodified and modified to test its efficiency for culture reproduction. Whey was deproteinised by ultrafiltration to remove particulate matter unnecessary for microbial metabolism, similar to the method carried out by Chessa et al. [2], where the use of clarified *scotta* (by centrifugation) as substrate allowed a higher cell concentration rate than unmodified *scotta*. However, in this case, ultrafiltration in WU substantially decreased protein content from 1 to 0.18%, resulting in a significantly lower concentration of several microbial groups (thermophilic cocci, lactobacilli, and mesophilic cocci). Therefore, WU was supplemented with bacteriological peptone (WU + P) to restore the original protein content. The unmodified whey (W) and the ultrafiltered whey supplemented with peptone (WU + P) resulted in the best substrates for NSC propagation since no variations in the concentrations of the microbial groups investigated were observed, i.e., thermophilic cocci and lactobacilli, mesophilic cocci and lactobacilli, enterococci, staphylococci, and coliforms (the latter two were not detected). The reproduction of a natural starter culture in standardised whey can have several advantages in managing the culture. First, propagation in whey is better than in milk when the culture needs to be concentrated and preserved by freezing or lyophilisation since centrifugation is not applicable to fermented milk due to coagulation. Moreover, after centrifugation, the concentration of NSC grown in whey turned out to be 20 times higher than that grown in milk. Furthermore, the use of whey for NSC reproduction can be a plus also in terms of waste recycling, since whey is a residual of cheesemaking that must be disposed of with additional costs for dairy farms [36]. The methodology described has limitations due to the fact that only a small portion of microorganisms can be cultured by current laboratory techniques, and the microbial diversity present in the NSC culture, like the whole microbial biodiversity present in the biosphere, is not completely known [37,38]. Thus, the raw milk microbial community cannot be completely kept in the NSC. However, the undefined culture obtained is reasonably more biodiverse than commercial starter cultures composed of a few species, each represented by a single strain, grown separately and then blended.

The loss of biodiversity in starter cultures can affect cheese flavours with repercussions on the cheese’s sensory properties. Moreover, the advantage of using natural cultures where microbial species are represented by several strains could avoid the loss of technological performance in acidification. This is supported by the “kill the winner” hypothesis, which assumes that bacteriophages infect the dominant bacterial strains [39]. Since the commercial starters are composed of a few strains, they can be more susceptible to bacteriophage infections [9]. Conversely, a natural culture represented by a multitude of strains can be more resilient. 

## 5. Conclusions

In the present study, a description of an innovative method to produce a natural starter culture that inhibits the growth of potentially pathogenic, low-hygienic-related and anti-dairy bacteria directly from milk without applying any heat treatment was described. The method aims to be applied to all milk types and stimulate the growth of thermophilic or mesophilic bacteria, depending on the incubation temperatures used. The purpose of this study was also to find a useful method to be applied both at the laboratory and industrial scale to produce a starter culture while safeguarding the microbial biodiversity of dairy plant environments and traditional, artisanal, and PDO products linked to the territory of production. Further studies will be needed before using the starter culture in cheesemaking to characterise the culture at the strain level and assure an adequate standard of safety for the non-QPS/GRAS strains that might constitute the undefined microbiota. 

## Figures and Tables

**Figure 1 microorganisms-11-00823-f001:**
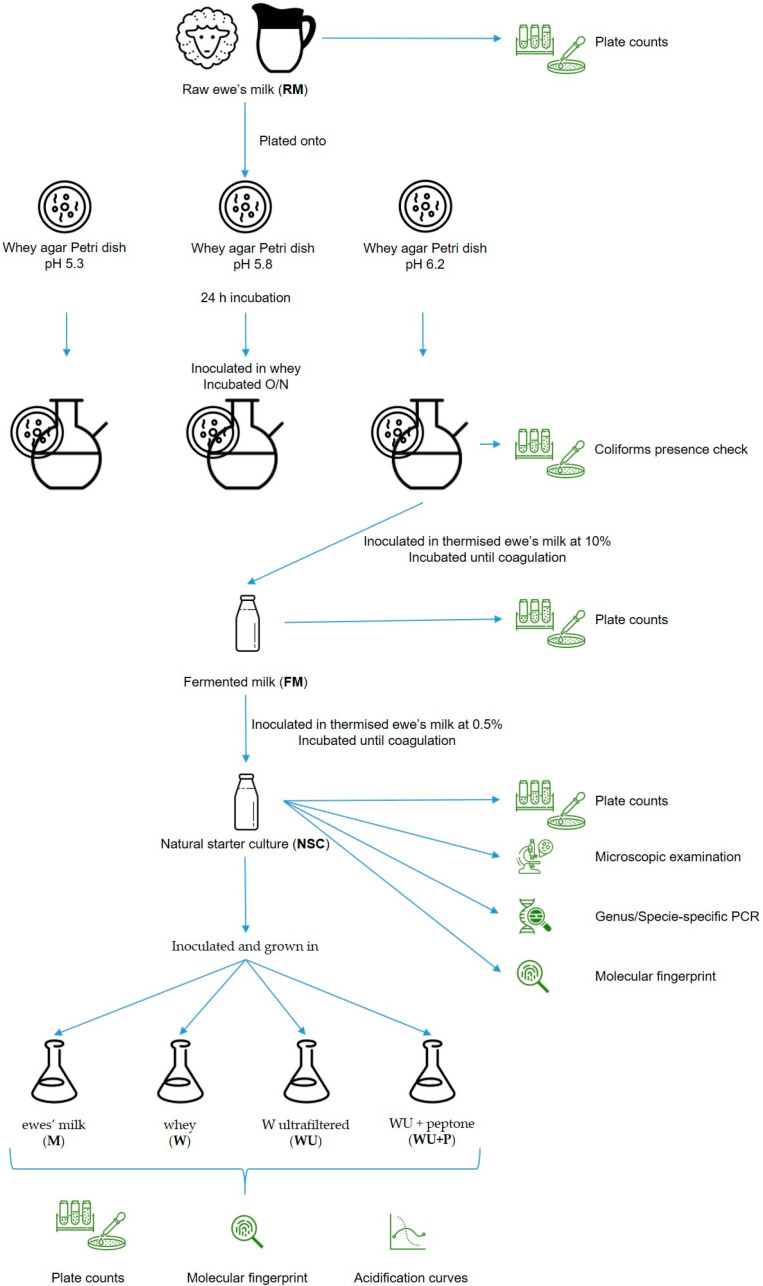
Experimental plan regarding the production of a natural starter culture (NSC). The raw ewe milk (RM) was plated onto whey agar dishes, each at a different pH, and, after incubation, transferred into a whey solution and incubated again for 24 h. Then, the whey inoculated with the plate at pH 6.2 was used to inoculate sterile ewes’ milk (at 10%) to obtain, after inoculation, fermented milk (FM). The FM microbiota was grown in sterile ewe milk to obtain NSC, which was characterised at the phenotypic and molecular scale. The development of NSC microbiota in milk (M), whey (W), W ultrafiltered (WU), and WU supplemented with peptone (WU + P) was also investigated and compared by plate count, molecular fingerprint, and kinetics of acidification.

**Figure 2 microorganisms-11-00823-f002:**
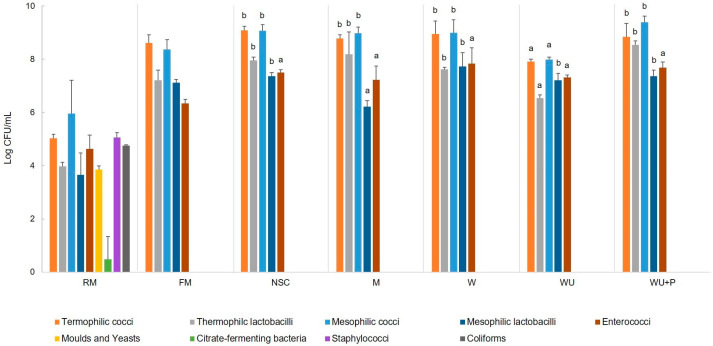
Microbial counts of thermophilic cocci and lactobacilli, mesophilic cocci and lactobacilli, enterococci, moulds and yeasts, citrate-fermenting bacteria, staphylococci, and coliforms in raw ewe milk (RM), fermented milk (FM), a natural starter culture (NSC), an NSC grown in ewe’s milk (M), whey (W), whey ultrafiltered (WU), and in whey ultrafiltered and supplemented with bacteriological peptone (WU + P). Microbial counts were expressed as Log CFU/mL ± standard deviation. For each microbial group detected in NSC, and NSC grown in M, W, WU, and WU + P, different letters above the respective column indicate significant (*P* < 0.05) differences in the Log CFU/mL, according to the Tukey–Kramer post hoc test.

**Figure 3 microorganisms-11-00823-f003:**
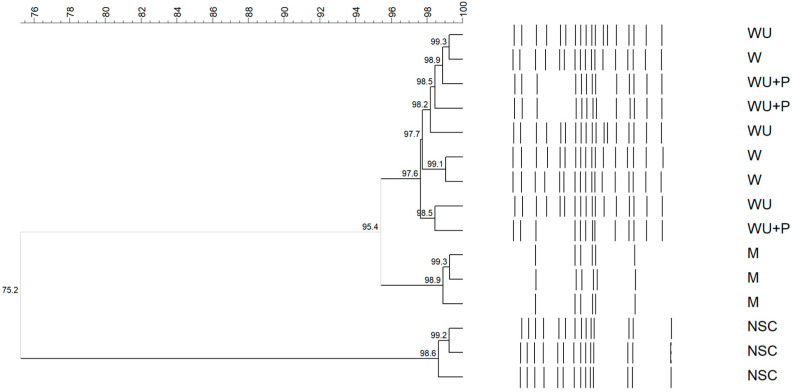
UPGMA cluster analysis, using Pearson correlation, of (GTG)5 rep-PCR fingerprints of total-community DNA extracted from the natural starter culture (NSC), NSC grown in ewe’s milk (M), in whey (W), in whey ultrafiltered (WU), and in whey ultrafiltered and supplemented with bacteriological peptone (WU + P). Three replicates for each sample were performed.

**Figure 4 microorganisms-11-00823-f004:**
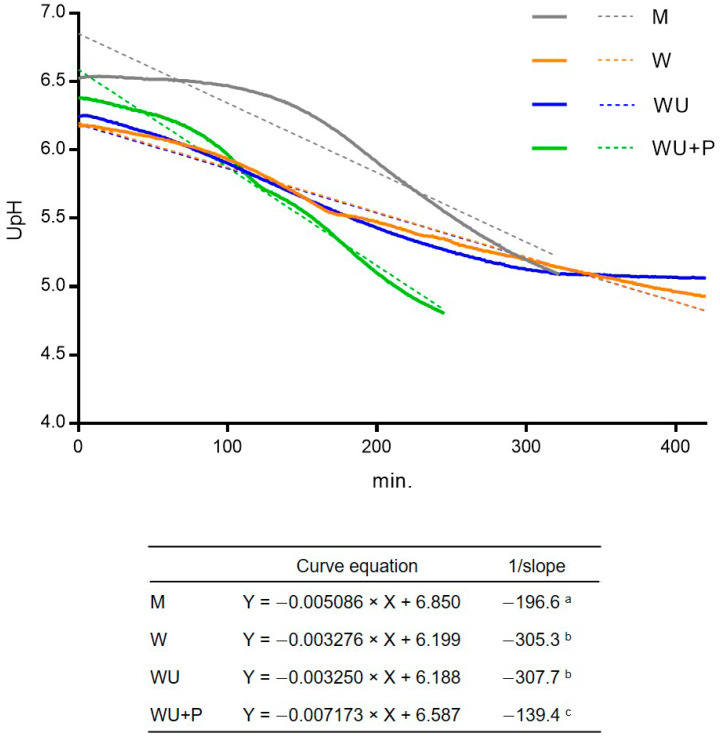
Acidification curves at 37 °C of NSC inoculated in ovine milk, whey (W), whey ultrafiltered (WU), and WU supplemented with bacteriological peptone (WU + P). Continuous lines indicate the acidification activity during incubation. Dashed lines indicate linear regression of the acidification curves. Curve equations and 1/slopes are indicated in the table at the bottom of the figure. The 1/slopes not sharing the same apex letter differ significantly (*P* < 0.0001), according to the Tukey-Kramer post hoc test.

**Table 1 microorganisms-11-00823-t001:** Primers used in this study for the detection of target genes and bacteria.

Target Bacteria	Target Gene	Primer	Primer Sequence (5′–3′)	Annealing (°C)	Size (bp)	Reference
*Lactococcus lactis*	*16S rRNA*	LcLspp-F	GTTGTATTAGCTAGTTGGTGAGGTAAA	55	387	[17]
		Lc-R	GTTGAGCCACTGCCTTTTAC			
*Lactobacillus delbrueckii* subsp *lactis*	*dppE*	Lac-LACTIS-F733	TGCCAAGCTCTACTCCGTTT	58	217	[18]
		Lac-LACTIS-R949	GTCAAGCGGCATAGTGTCAA			
*Lactobacillus bulgaricus*	*lacZ*	Lac-BULG-F391	GGAAGACTCCGTTTTGGTCA	58	395	[18]
		Lac-BULG-R785	AGTTCAAGTCTGCCCCATTG			
*Lactobacillus helveticus*	*prtH*	Lac-HELV-F73	GGCGGGGAAAGAGGTAACTA	58	509	[18]
		Lac-HELV-R581	TGACGCAAACTTAATGAACCA			
*Streptococcus thermophilus*	*lacZ*	Str-THER-F2116	GCTTGTGTTCTGAGGGAAGC	58	577	[18]
		Str-THER-R2693	CTTTCTTCTGCACCGTATCCA			
*Limosilactobacillus fermentum*	*arcD*	Lac-FER-F753	CCAGATCAGCCAACTTCACA	58	310	[18]
		Lac-FER-R1062	GGCAAACTTCAAGAGGACCA			
*Lactobacillus reuteri*	*16S rRNA*	REUT1	TGAATTGACGATGGATCACCAGTG	65	1000	[19]
		LOWLAC	CGACGACCATGAACCACCTGT			
*Lacticaseibacillus paracasei*	*16S rRNA*	Y2	CCCACTGCTGCCTCCCGTAGGAGT	55	290	[20]
		PARA	CACCGAGATTCAACATGG			
*Lactiplantibacillus plantarum*	*recA*	planF	CCGTTTATGCGGAACACCTA	56	318	[21]
		pREV	TCGGGATTACCAAACATCAC			
*Enterococcus* spp.	*tuf*	ENT1	TACTGACAAACCATTCATGATG	59	112	[22]
		ENT2	AACTTCGTCACCAACGCGAAC			
*Enterococcus faecium*	*sodA*	FM1	GAAAAAACAATAGAAGAATTAT	55	215	[23]
		FM2	TGCTTTTTTGAATTCTTCTTTA			
*Enterococcus faecalis*	*sodA*	FL1	ACTTATGTGACTAACTTAACC	55	360	[23]
		FL2	TAATGGTGAATCTTGGTTTGG			
*Streprococcus gallolyticus* subsp. *macedonicus*	*16S rRNA*	16MAC	TAGTGTTTAACACATGTTAGAGA	57	350	[24]
		BSR534/18	ATTACCGCGGCTGCTGGC			

## Data Availability

The data presented in this study are available on request from the corresponding author.

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
