# Peer review of "A Strategy for the Recovery of Raw Ewe’s Milk Microbiodiversity to Develop Natural Starter Cultures for Traditional Foods"

_microorganisms, 2023, doi:10.3390/microorganisms11040823_

Round 1

Reviewer 1 Report

The manuscript “A strategy for the recovery of raw ewe's milk micro biodiversity to develop natural starter cultures for traditional foods” is an interesting and important study that aims to preserve the natural starter cultures from raw ewe’s milk by removing potentially harmful microorganisms. The study is well-written, the methodology is comprehensive, and it has a good flow.

Some corrections should be implemented, and please revise that every statement is properly cited in the whole manuscript.

-          please define the limitations of the study

Abbreviations should be described at first use, in the abstract (line 13) and in the whole text (line 44) also, as they are considered separate texts. Revise the whole manuscript for the correct usage of abbreviations.

Some minor corrections, revise the whole manuscript - this is only highlighted for the abstract - revise for correctness:

line 8 – on the basis of > based on

line 12 - most typical and PDO food products are > most typical PDO food products are

line 14 – be not > not be

line 15 – pathogen > pathogens

line 323 – “This study is a continuum of previous research conducted on natural starter cultures about their” - please cite the previous researches

lines 325-328 – who did the “were reproduced and propagated in the natural medium” – please define the reference

The manuscript is well written, with significant data sufficiently described and discussed. After some thorough revisions, it could be considered for publication.

Reviewer 2 Report

The manuscript describes a new procedure to prepare natural starter cultures for food fermentation. The issue of natural starters is important since, still and especially in Italy, many cheeses are made with them. It is also true that working with these cultures is not easy since their complex microbiological composition depends on many factors (type of milk, geography, time of year, preparation technology, etc.). They offer important advantages, which are well mentioned by the authors, but they also have severe disadvantages due to the instability of the microbial diversity they contain (which I attest to), which means that sooner or later they lose, for example, their acidification speed. and it is necessary to continue producing, look elsewhere for a well-functioning culture, or start making a new culture.

The idea of designing a new methodology to prepare a natural culture from sheep's whey that can preserve the microbial diversity of the starting milk is a good one. This intention is highly valued, but the presentation of the method as it appears in the manuscript leaves many doubts and has important omissions, which are detailed below.

- The genus and species of microorganisms must always be written in italics. See References

2.2 - This point has several problems:

-line 104: were the 2 milkings mixed or processed separately? Were they from the same day?

- line 112: powdered sheep whey?

- line 116: was 1 ml of sample plated? On the surface or poured in the agar plate? If plating was on the surface, 1 ml is too much for the agar layer to absorb.

- lines 118 and 227: how was it transferred? Was the entire plate transferred or was the surface washed to wash away bacterial growth?

-- line 293: delete "(Figure 3)". It already appears on line 290

- line 135: what does "three distinct replicates" mean? 3 replicates of the entire procedure with the same milk sample?

Fig. 1 has several problems. Some process steps listed in 2.2 are missing. If it is intended to show what is indicated in 2.2, complete it. If not, skip it

- RM counts: were they made on the sample as soon as it arrived at the laboratory (Fig. 1) or after freezing/thawing (lines 108-110)? The ideal would be as soon as it arrives. Are the authors sure that freezing at -80ºC, even though cryoprotectant is used, does not modify the complex diversity of RM? They should prove it given that they frequently use freezing to maintain samples with high and diverse microbiota

- To what do the authors attribute the rapid disappearance of coliforms and staphylococci, since the pHs to which the Whey Agar adjusts are not too low to inhibit them? Moreover, the highest ph is adopted (6.2)

- The microbiota undergoes 4 multiplications (there are 4 incubations) (lines 117, 119, 122 and 130), to which another cell multiplication should be added during cheese making. It is risky to think that, regardless of the counts achieved, the diversity of the lactic microbiota present in RM can be maintained and be beneficial for cheese production.

- Comment on the idea of the study: NSC was obtained from 1 RM sample. Going from the laboratory to the industrial reality, how should it be done so that an industry has NSC to make the same cheese over time? Use the frozen aliquots? And if the starting RM does not have the appropriate microbiota?

Round 2

Reviewer 1 Report

The authors implemented all the required corrections in the manuscript. It can be accepted for publication.

Reviewer 2 Report

The authors have sufficiently clarified the doubts that arose from reading the first version and the current manuscript is an improved version, suitable for publication.